# Evaluation of the Cytotoxicity and Oxidative Stress Response of CeO_2_-RGO Nanocomposites in Human Lung Epithelial A549 Cells

**DOI:** 10.3390/nano9121709

**Published:** 2019-11-29

**Authors:** Maqusood Ahamed, Mohd Javed Akhtar, M. A. Majeed Khan, ZabnAllah M. Alaizeri, Hisham A. Alhadlaq

**Affiliations:** 1King Abdullah Institute for Nanotechnology, King Saud University, Riyadh 11451, Saudi Arabia; mohd.j.akhtar@gmail.com (M.J.A.); majeed_phys@rediffmail.com (M.A.M.K.); hhadlaq@ksu.edu.sa (H.A.A.); 2Department of Physics and Astronomy, College of Science, King Saud University, Riyadh 11451, Saudi Arabia; zabn1434@gmail.com

**Keywords:** CeO_2_-RGO nanocomposite, biocompatibility, cytotoxicity, oxidative stress, biomedical applications

## Abstract

Graphene-based nanocomposites have attracted enormous interest in nanomedicine and environmental remediation, owing to their unique characteristics. The increased production and widespread application of these nanocomposites might raise concern about their adverse health effects. In this study, for the first time, we examine the cytotoxicity and oxidative stress response of a relatively new nanocomposite of cerium oxide-reduced graphene oxide (CeO_2_-RGO) in human lung epithelial (A549) cells. CeO_2_-RGO nanocomposites and RGO were prepared by a simple hydrothermal method and characterized by relevant analytical techniques. Cytotoxicity data have shown that RGO significantly induces toxicity in A549 cells, evident by cell viability reduction, membrane damage, cell cycle arrest, and mitochondrial membrane potential loss. However, CeO_2_-RGO nanocomposites did not cause statistically significant toxicity as compared to a control. We further observed that RGO significantly induces reactive oxygen species generation and reduces glutathione levels. However, CeO_2_-RGO nanocomposites did not induce oxidative stress in A549 cells. Interestingly, we observed that CeO_2_ nanoparticles (NPs) alone significantly increase glutathione (GSH) levels in A549 cells as compared to a control. The GSH replenishing potential of CeO_2_ nanoparticles could be one of the possible reasons for the biocompatible nature of CeO_2_-RGO nanocomposites. Our data warrant further and more advanced research to explore the biocompatibility/safety mechanisms of CeO_2_-RGO nanocomposites in different cell lines and animal models.

## 1. Introduction

Graphene is a single atom thick two-dimensional sheet of sp^2^ hybridized carbon atoms with multifaceted properties, such as elasticity, flexibility, high conductivity, and ease of functionalization [1,2]. Due to these properties, graphene has attracted enormous interest in synthetic biology and nanomedicine, besides its applications in photonics, optoelectronics, and environmental remediation [3,4,5]. Despite the great potential of graphene in biomedical applications, its unmodified state has poor solubility in biological media, which makes nanomedicine study in this context complicated [6,7]. To overcome this problem, one of the most successful approaches has been the use of graphene derivatives such as graphene oxide (GO) and reduced graphene oxide (RGO). GO can be prepared by the oxidation of graphite, and RGO is the reduced form of GO [8]. GO and RGO exhibit excellent dispersibility and stability in physiological media because of their oxygenated functional groups [9].

The effects of GO and RGO on humans and environmental health are presently being investigated. Apart from plenty of literature on the exponential biomedical applications, a limited and contrasting amount of results are available on biocompatibility/toxicity of GO and RGO. For example, Wang et al. [10] observed that after intravenous injection, GO was mainly located in the lung, liver, and spleen, causing some level of toxicity to these tissues. However, another in vivo study found no toxicity after three months of GO injection in mice [11]. Some in vitro studies have reported the biocompatible nature of GO and RGO [9,12,13]. However, several in vitro reports have shown that GO and RGO can induce significant toxicity to various cell lines, such as stem cells, germ cells, lung cells, skin cells, endothelial cells, and macrophages [14,15,16,17,18,19]. Conflicting results on the toxicity/biocompatibility of graphene derivatives could be due to different preparation methods and the manipulation of physicochemical properties.

Cerium oxide (CeO_2_) is an excellent semiconductor oxide, owing to its wide band gap (~3.19 eV). CeO_2_ nanoparticles (NPs) has spurred interest in the biomedical field because of its antioxidative and anti-inflammatory properties [20,21]. Currently, researchers are synthesizing GO/RGO-based nanocomposites with better optoelectronic properties than those of pure GO or RGO. For example, Fe_3_O_4_-RGO and CeO_2_-RGO nanocomposites display improved an adsorption behavior and photocatalytic activity than pure RGO [22,23]. The improved optoelectronic properties of these nanocomposites make them substantial candidates for tissue imaging, biosensing, and targeted drug delivery. Diaz-Diestra et al. [24] prepared non-toxic folic acid functionalized RGO/ZnS:Mn nanocomposites that show potential to be used as a platform for targeted cancer treatment. Hence, it is necessary to thoroughly examine the biological interaction of these graphene-based nanocomposites before their applications in nanomedicine and environmental remediation. Studies on the biological response of metal oxide-based RGO nanocomposites at the cellular and molecular levels are scarce.

This is the first study that explores the cytotoxic and apoptotic potential of CeO_2_-RGO nanocomposites in human lung cancer epithelial (A549) cells. The cytotoxicity mechanisms of CeO_2_-RGO nanocomposites are examined through oxidative stress. Lung cancer cell lines are commonly utilized to examine the cytotoxicity of an experimental material for biomedical applications. Cancer cell lines such as the lung were selected by National Cancer Institute as a model to screen drugs/compounds as a prelude to testing in xenografts or animal models [25]. Moreover, the widespread applications of these nanocomposites may increase their chance of exposure to human lung cells. In this regard, A549 cells have been widely used by toxicology and pharmacology researchers [26]. CeO_2_-RGO nanocomposites were prepared by a simple hydrothermal method. The characterization of CeO_2_-RGO nanocomposites was done by X-ray diffraction (XRD), X-ray photoelectron spectroscopy (XPS), scanning electron microscopy (SEM) and transmission electron microscopy (TEM). Information on the physicochemical properties of CeO_2_-RGO nanocomposites and their cellular response with possible toxicity mechanisms are critically important to study, in order to outline the potential application of this nanocomposite in the field of biomedicine.

## 2. Materials and Methods

### 2.1. Preparation of CeO_2_-RGO Nanocomposites, RGO and CeO_2_ NPs

Graphene oxide (GO) was prepared by an improved Hummer’s procedure, utilizing pure graphite powder [27,28]. The CeO_2_-RGO nanocomposites were prepared by a facile hydrothermal method using GO and cerium chloride (CeCl_3_ 7H_2_O) as substrates [29]. Also, RGO was synthesized without mixing the of CeCl_3 _7H_2_O. For a comparative study, CeO_2_ NPs were also prepared without adding GO, according to the protocol of Kumar and Kumar [29].

### 2.2. Characterization

The phase purity and crystal structure of CeO_2_ NPs, RGO, and CeO_2_-RGO nanocomposites were examined by a X-ray diffractometer (XRD, PanAnalytic X’Pert Pro) employing Cu-Kα radiation (λ = 0.15405 nm, at 45 kV and 40 mA). The surface chemical composition of CeO_2_-RGO nanocomposites was assessed by X-ray photoelectron spectroscopy (XPS) (PHI-5300 ESCA PerkinElmer, Boston, MA, USA). The morphologies of the CeO_2_ NPs, RGO and CeO_2_-RGO nanocomposites were determined by scanning electron microscopy (SEM, JSM-7600F, JEOL, Inc., Tokyo, Japan) and transmission electron microscopy (TEM, JEM-2100F, JEOL, Inc., Tokyo, Japan).

### 2.3. Cell Culture

Human lung epithelial (A549) cells were cultured in Dulbecco’s modified eagle’s medium (DMEM) (Invitrogen, Carisbad, CA, USA) with the supplementation of 100 units/mL of pencillin and 100 µg/mL of streptomycin antibiotics (Invitrogen), along with 10% fetal bovine serum (Invitrogen) at 37 °C, with a 5% CO_2_ supply. At 80% confluence, cells were harvested and further cultured for toxicity experiments.

### 2.4. Exposure of Cells to CeO_2_ NPs, RGO and CeO_2_-RGO Nanocomposites

Initially, we performed 3-(4,5-Dimethylthiazol-2-yl)-2,5-diphenyltetrazolium bromide (MTT) screening tests to choose an appropriate concentration for the CeO_2_ NPs, RGO, and CeO_2_-RGO nanocomposites. Cells were exposed to different concentrations (1, 5, 10, 25, 50, and 100 µg/mL) of CeO_2_ NPs, RGO, and CeO_2_-RGO nanocomposites for 24 h. On the basis of screening tests, one concentration (50 µg/mL) of each material (CeO_2_ NPs, RGO, and CeO_2_-RGO nanocomposites) was used to explore the toxicity mechanisms.

### 2.5. Toxicity Experiments

Cells were treated for 24 h to CeO_2_ NPs, RGO and CeO_2_-RGO nanocomposites at a concentration of 50 µg/mL. Cell viability was measured by MTT and neutral red uptake (NRU) assays, with some specific changes [30]. The MTT assay is based the principle that live cells have the ability to reduce MTT in blue formazan products that are dissolved in a solvent. The absorbance of this product was measured at 570 nm using a microplate reader (Synergy-HT Biotek, Winooski, VT, USA). In the NRU assay, live cells have ability to bind neutral red (NR) dye with lysosomes. The absorbance of the NR probe was determined at 540 nm using a microplate reader (Synergy-HT, Biotek). The lactate dehydrogenase (LDH) enzyme has the ability to oxidize lactate into pyruvate. The LDH enzyme resides in cytoplasm and leaks into a culture medium when the cell membrane is damaged. LDH leakage was measured by a BioVision kit (Milpitas, CA, USA), and the detailed procedure here has been reported in our earlier paper [31]. A propidium iodide (PI) probe was applied to determine the cell cycle phases using a flow cytometer (Coulter Epics XL/XI-MCL) via FL2 filter (585 nm) [32]. Caspase-3 enzyme activity was measured using a BioVision colorimetric kit. Mitochondrial membrane potential (MMP) was examined by a rhodamine-123 (Rh-123) probe [32]. The cationic fluorochrome Rh-123 binds to mitochondria of live cells in a membrane potential-dependent manner. The MMP level was measured by two different methods, namely, quantitative analysis by a microplate reader (Synergy-HT, Biotek) and qualitative examination by a fluorescent microscope (DMi8, Leica Microsystems, Wetzlar, Germany). A dichlorofluorescin diacetate (DCFH-DA) probe was employed to assess the intracellular level of reactive oxygen species (ROS), as reported earlier [32]. DCFH-DA passively enters the cells and reacts with ROS to form a fluorescent compound called dichlorofluorescein (DCF). The fluorescent intensity of DCF was determined by two different methods, namely, quantitative analysis by a microplate reader (Synergy-HT, Biotek) and qualitative assessment by a fluorescent microscope (DMi8, Leica Microsystems, Germany). Hydrogen peroxide (H_2_O_2_) generation in cells was measured by a commercial kit (Sigma-Aldrich, St. Louis, Missouri, USA). The glutathione (GSH) level was estimated by Ellman’s procedure, using 5,5-dithio-bis-nitrobenzoic acid (DTNB) [33]. Protein content was estimated by Bradford’s method [34].

### 2.6. Statistical Analysis

A one-way analysis of variance (ANOVA), followed by Dunnett’s multiple comparison test, was employed for the statistical analysis of the toxicity results. Here, *p* < 0.05 was assigned as statistical significance.

## 3. Results and Discussion

### 3.1. Characterization of CeO_2_ NPs, RGO and CeO_2_-RGO Nanocomposites

To examine the crystal structure of the prepared samples, XRD patterns of the RGO, CeO_2_ NPs, and CeO_2_-RGO nanocomposites were recorded (Figure 1). The XRD profile of RGO demonstrated an intense reflection plane (002) at 2θ = 24.92. The presence of another diffraction peak at 2θ = 42.97, which is attributed to the (100) plane, suggests the polycrystalline nature of the prepared RGO. The diffraction peaks in the XRD pattern of the prepared CeO_2_ NPs were also recorded at 2θ values of 28.87°, 33.32°, 47.68°, 56.57°, 59.37°, and 69.79°, which correspond to the (111), (200), (220), (311), (222) and (400) crystal planes, respectively. All the peaks were indexed to the pure cubic fluorite-type structure of CeO_2_ (JCPDS: 75-0076) [35]. In the XRD pattern of the CeO_2_-RGO nanocomposite, two additional peaks of RGO [(002) and (100)] were also found with all the peaks of pure CeO_2_ NPs (Figure 1). These results indicated that CeO_2_ NPs were fairly anchored on RGO nanosheets. The average crystallite sizes of CeO_2_ NPs and CeO_2_-RGO nanocomposites were estimated by Scherrer’s formula, using the prominent peak (111), and it was found to be around 18 and 16 nm for CeO_2_ NPs and CeO_2_-RGO nanocomposites, respectively. Our XRD data on RGO, CeO_2_ NPs, and CeO_2_-RGO nanocomposites were according to earlier reports [36,37].

X-ray photoelectron spectroscopy (XPS) was done to further analyze the chemical composition of CeO_2_-RGO nanocomposites. The presence of C (C1s), O (O1s), and Ce (Ce 3d) elements in the XPS analysis suggests that the CeO_2_-RGO nanocomposite was successfully prepared. The peaks that appeared at 888.9, 895.1, and 903.9 eV correspond to the components of Ce3d5/2, while the remaining signals of Ce3d3/2 components could be seen at 906.9, 913.6 and 923.1 eV (Figure 2A) [35,38]. The XPS spectra of C1s given in Figure 2B contain a non-oxygenated aromatic sp^2^ bonded carbon ring (C-C) at 286.5 eV [39]. Figure 2C represents the O1s spectra of CeO_2_, with two intense peaks at 530.9 and 536.8 eV, linked with anionic oxygen in CeO_2_ and the presence of the residual oxygen functional group in the RGO sheets, respectively [37].

The morphologies of the CeO_2_ NPs, RGO, and CeO_2_-RGO nanocomposites were further characterized by SEM and TEM (Figure 3A–F). Figure 2A,D represents the TEM and SEM morphology of CeO_2_ NPs, which show some level of agglomeration. The SEM and TEM images of RGO (Figure 2B,E) indicate the formation of few layers of RGO, with visible wrinkles and a silky morphology because of their high aspect ratio. The SEM and TEM micrographs of CeO_2_-RGO nanocomposites (Figure 2C,F) indicate that the CeO_2_ NPs were well anchored on the sheets of RGO. It is also noticeable from the SEM and TEM images that the CeO_2_-RGO nanocomposites still maintained layered structures, although some distinct wrinkles disappeared [36].

### 3.2. Cytotoxicity

As we can see in Figure 4A, the CeO_2_ NPs were not toxic to A549 cells up to the concentration of 100 µg/mL. The non-cytotoxic nature of CeO_2_ NPs against human breast cancer (MCF-7) and human fibroblast (HT1080) cells was also observed in our previous work [21]. However, RGO was found to induce dose-dependent cytotoxicity in A549 cells in the concentration range of 25–100 µg/mL (Figure 4B). Below the concentration of 25 µg/mL, RGO was also non-cytotoxic. Our results were according to other recent reports that demonstrated the cytotoxic potential of RGO in different types of human cells [6,7]. However, Bengtson et al. [13] reported that GO and RGO did not produce cytotoxicity and genotoxicity to murine lung epithelial (FE1) cells. These contrasting results could be due to the use of different cell lines, different preparation methods, and the manipulation of physicochemical properties. Interestingly, CeO_2_-RGO nanocomposites did not induce cytotoxicity to A549 cells in all selected concentration ranges (1–100 µg/mL) (Figure 4C).

The toxicity of GO and RGO has been observed in different types of cells, including human breast cancer (MCF-7), liver cancer, and (HepG2) and lung (A549) cells, all with significant cytotoxicity detected at a concentration of around 50 µg/mL [10,12]. Hence, we have chosen one concentration (50 µg/mL) of CeO_2_ NPs, RGO, and CeO_2_-RGO nanocomposites for further experiments to explore the toxicity mechanisms of these nanoscale materials. The A549 cells were exposed for 24 h to 50 µg/mL of CeO_2_ NPs, RGO, and CeO_2_-RGO nanocomposites, and cytotoxicity was assessed by MTT, NRU, and LDH assays. As we can see in Figure 5A,B, the CeO_2_ NPs did not reduce cell viability. However, cell viability was significantly decreased in the RGO-treated group as compared to the control group. Interestingly, compared to the RGO group, in the CeO_2_-RGO nanocomposites group, cell viability was significantly increased to the level that was not much less than the CeO_2_ NPs or control groups (Figure 5A,B). Cytosolic enzyme LDH leakage into the culture medium is an indicator of cell membrane damage. Our results demonstrated that CeO_2_ NPs did not cause LDH leakage, however RGO significantly induced LDH leakage in the culture medium of A549 cells. Besides, compared to RGO group, in the CeO_2_-RGO nanocomposite group, leakage of the LDH enzyme was significantly reduced to a level that was almost close to the CeO_2_ NPs or the control group (Figure 5C). These results indicated that CeO_2_-RGO nanocomposites induce much less cytotoxicity to A549 cells then RGO.

### 3.3. Apoptosis

Apoptotic cell death is a highly regulated process for the removal of irreparable damaged cells to maintain tissue homeostasis [40]. The process of apoptosis can be triggered by several factors, including nutrient deficiency, growth factors, and environmental contaminants [41]. GO and RGO have shown potential to induce apoptosis in human cells [7,18,19,26]. Ali et al. [18] also reported that Ag-doped RGO triggered apoptosis in human hepatic normal and carcinoma cells. Hence, we further examined the apoptotic potential of the prepared materials in A549 cells. Cell cycle phases were analyzed in A549 cells after exposure for 24 h to 50 µg/mL of the CeO_2_ NPs, RGO, and CeO_2_-RGO nanocomposites. Cells with damaged DNA are accumulated in the G1 (gap1), S (DNA synthesis, or in G2/M (gap2/mitosis) phases for DNA repair. However, cells with irreparable DNA damage are directed to undergo apoptosis and are anticipated to accumulate in the sub-G1 phase [42]. Our flow cytometer data indicated that RGO induced apoptosis in A549 cells, however, the apoptotic response of the CeO_2_-RGO nanocomposites was indeed much lower than RGO. The CeO_2_ NPs also did not induce apoptosis in A549 cells. Cells accumulated in all phases of the cell cycle were almost similar in the control and CeO_2_ NPs groups. However, cell accumulation in sub-G1 phase of the RGO group was significantly higher (6.88%) than those of the control group (4.16%) (*p* < 0.05) (Figure 6A). Interestingly, the number of cells accumulated in the sub-G1 phase in CeO_2_-RGO nanocomposites was much lower (4.32%) than the RGO group (6.88%) and was almost similar to the control group (4.16%) (Figure 6A). The caspase-3 enzyme also plays a significant role in apoptotic pathway. Our results demonstrate that the activity of caspase-3 enzyme is similar in both the CeO_2_ NPs and the control group. However, caspase-3 enzyme activity was significantly higher in the RGO group than that of control group. Interestingly, we observed that the activity of the caspase-3 enzyme in the CeO_2_-RGO nanocomposite group was much lower than in RGO and was almost similar to the control group (Figure 6B).

The mitochondrial membrane potential (MMP) level is a hallmark of apoptosis [43]. MMP loss is also an indicator that cells are going to die through the apoptotic pathway. Hence, we further examined the MMP level in A549 cells against CeO_2_ NPs, RGO, and CeO_2_-RGO nanocomposites exposure at a concentration of 50 µg/mL for 24 h. The quantitative data demonstrated that the MMP level in CeO_2_ NPs and control group was similar. However, in RGO group, MMP level was significantly lower as compared to the control group. In addition, the MMP level in CeO_2_-RGO nanocomposites were much higher than those of the RGO group and were almost close to the control group (Figure 7A). Fluorescent microscopy data also suggested that the brightness of the Rh-123 probe was almost similar in the CeO_2_ NPs, CeO_2_-RGO nanocomposite, and control groups, while much lower in the RGO group (Figure 7B). These results suggest that the CeO_2_-RGO nanocomposite induced much less of a apoptotic response in A549 cells than those of the RGO nanosheets.

### 3.4. Oxidative Stress

The underlying mechanisms of nanoscale materials toxicity are not completely delineated. Oxidative stress generation, through which nanoscale materials exert toxicity, is now being studied by investigators [44]. Nanoscale materials might induce oxidative stress, either by the generation of pro-oxidants or by the reduction of antioxidants. A higher production of intracellular ROS or the depletion of antioxidants may lead to the oxidative damage of cell biomolecules [45]. ROS such as the superoxide anion (O_2_^•−^), hydroxyl radical (HO•), and hydrogen peroxide (H_2_O_2_) serve as signaling molecules in the process of apoptosis [46]. The GO-ZnO nanocomposite was found to induce oxidative stress mediated toxicity in MCF-7 cells [47]. In this study, we examined several parameters of oxidative stress in A549 cells against CeO_2_ NPs, RGO, and CeO_2_-RGO nanocomposite when exposed to said particles for 24 h. The quantitative data demonstrated that the intracellular level of ROS in the CeO_2_ NP group was almost similar to that of the control group, but significantly higher than the RGO group (Figure 8A). Interestingly, in the CeO_2_-RGO nanocomposite group, the ROS level was significantly lower than the RGO group and was not statistically different from the control group. Similar to the quantitative data, fluorescent microscopy images demonstrated that the brightness of the DCF probe (marker of ROS level) in CeO_2_ NPs and the CeO_2_-RGO nanocomposite was almost close to the control group, but the brightness of DCF in the RGO group was much higher than that of the control group (Figure 8B). RGO group also showed low cell density because several cells died in 24 h exposure time of RGO. Besides, we assessed the H_2_O_2_ level in A549 cells against CeO_2_ NPs, RGO, and CeO_2_-RGO nanocomposite exposure for 24 h. As we can see in Figure 8C, the H_2_O_2_ level in the CeO_2_ NP group was not different from the control group. However, the H_2_O_2_ level in RGO was significantly higher in comparison to the control and CeO_2_ NP groups. Again, an interesting result was that the H_2_O_2_ level in the CeO_2_-RGO nanocomposites was significantly lower than the RGO group and was not statistically different from the control group (Figure 8C). In similarity, Qiang et al. [26] observed that Pb^2+^ reduced the cytotoxicity, oxidative stress, and apoptotic response of GO through the regulation of the GO morphology.

The equilibrium between pro-oxidant generation and their scavenging by antioxidant molecules and enzymes is a delicate phenomenon. The antioxidant molecule glutathione (GSH) plays an important role in cellular defense against oxidant injury. Several redox modulating enzymes, including thiol reductases and peroxidase, rely on the intracellular GSH pool as their source of reducing equivalents [48]. Intracellular GSH depletion has also been linked with either the induction or stimulation of apoptosis [49]. In this study, we observed that RGO significantly decreased the GSH level as compared to the control group. Furthermore, the GSH levels in CeO_2_-RGO nanocomposites were significantly higher than in the RGO group and were not statically different from the control group (Figure 8D). Here, we found a very interesting result, namely, that CeO_2_ NPs significantly increase GSH levels in A549 cells as compared to the control. The GSH replenishing potential of CeO_2_ NPs was also reported in our previous work [21]. Some other studies have also reported the antioxidative potential of CeO_2_ NPs [50,51]. This could be one of the possible reasons behind the biocompatible nature CeO_2_-RGO nanocomposite to A549 cells as compared to pure RGO. This study highlights the role GSH in the replenishing potential of CeO_2_ NPs when converting toxic RGO into biocompatible CeO_2_-RGO nanocomposites. However, the elucidating the underlying mechanisms through which CeO_2_ NPs convert cytotoxic RGO nanosheets into biocompatible CeO_2_-RGO nanocomposites still remains a daunting task.

Altogether, our data demonstrate that RGO induces toxicity in A549 cells. However, CeO_2_-RGO nanocomposites showed good biocompatibility toward A549 cells. The biocompatible nature of CeO_2_-RGO nanocomposites can be utilized for their potent application in a number of biomedical fields, such as drug delivery and biosensing. Earlier studies have also suggested the biomedical application of graphene-based nanocomposites. For instance, Wang et al. [52] observed that gold (Au)-RGO nanocomposites can effectively carry the anticancer drug doxorubicin inside human liver cancer (HepG2) cells. Au-RGO nanocomposites can also be used for the detection of micro-RNA [53]. A recent review suggested that Ag-RGO and Au-RGO nanocomposites have shown potential to be applied in drug delivery, cancer therapy, and biosensors [54].

## 4. Conclusions

CeO_2_-RGO nanocomposites and RGO were prepared and characterized by XRD, XPS, SEM, and TEM. The cytotoxic and apoptotic responses of RGO and CeO_2_-RGO nanocomposites were explored in A549 cells. We observed that RGO significantly induce cytotoxicity (cell viability reduction and LDH leakage), apoptosis (cell cycle arrest and MMP loss), and oxidative stress (ROS generation and GSH depletion) in A549 cells. However, CeO_2_-RGO nanocomposites did not cause a statistically significant change in toxicity to A549 cells as compared to the control group. Oxidative stress experiments suggested that the GSH replenishing potential of CeO_2_ NPs could be one of the potential mechanisms for the biocompatible nature the CeO_2_-RGO nanocomposite. Our encouraging data suggest that CeO_2_-RGO nanocomposites are a safe material at the cellular level. This study further provokes more advanced research for the development of graphene-based nanocomposites for their application in biomedicine.

## Figures and Tables

**Figure 1 nanomaterials-09-01709-f001:**
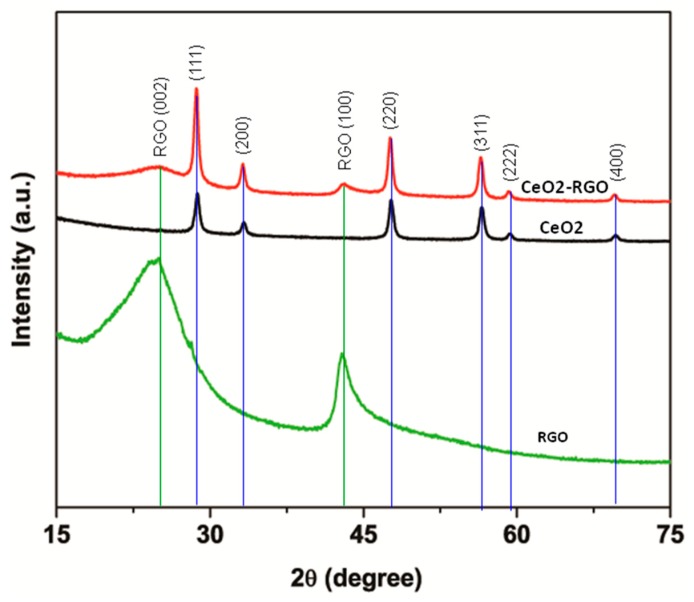
XRD characterization of CeO_2_ NPs, RGO and CeO_2_-RGO nanocomposites. XRD: X-ray diffraction; CeO_2_: Cerium oxide; NPs: Nanoparticles; RGO: Reduced graphene oxide.

**Figure 2 nanomaterials-09-01709-f002:**
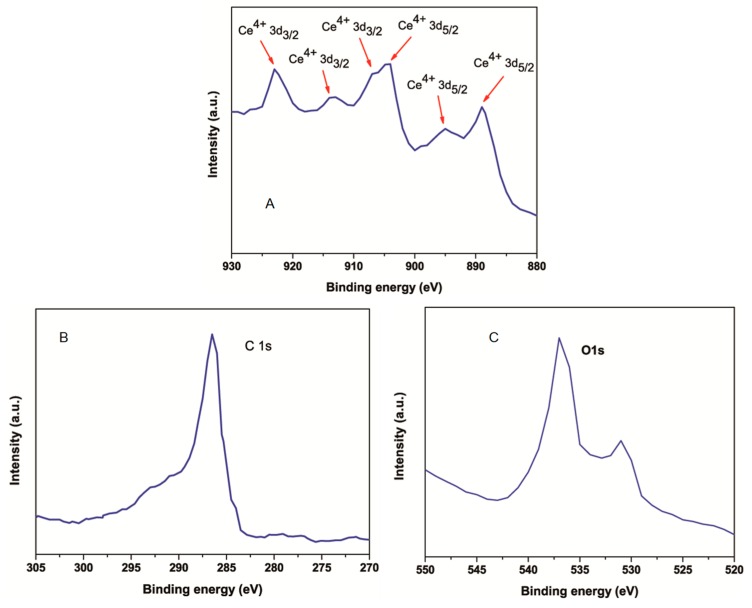
XPS analysis of CeO_2_-RGO nanocomposites. (**A**) Ce3d spectra, (**B**) C1s spectra, and (**C**) O1s spectra of CeO_2_-RGO nanocomposites. XPS: X-ray photoelectron spectroscopy; CeO_2_: Cerium oxide; NPs: Nanoparticles; RGO: Reduced graphene oxide.

**Figure 3 nanomaterials-09-01709-f003:**
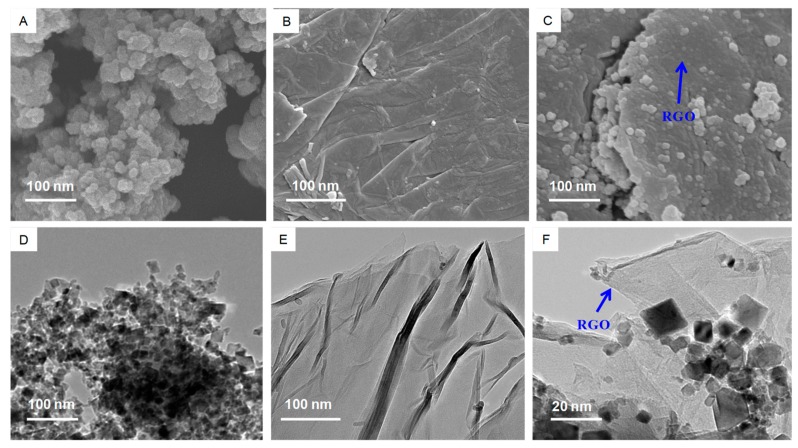
Electron microscopy characterization of RGO and CeO_2_-RGO nanocomposite. (**A**) SEM image of RGO. (**B**) SEM image of CeO_2_-RGO nanocomposite. (**C**) TEM image of RGO. (**D**) TEM image of CeO_2_-RGO nanocomposite. (**E**) TEM image of RGO. (**F**) TEM image of –CeO_2_-RGO nanocomposites. RGO: Reduced graphene oxide; CeO_2_: Cerium oxide; SEM: Scanning electron microscopy; TEM: Transmission electron microscopy.

**Figure 4 nanomaterials-09-01709-f004:**
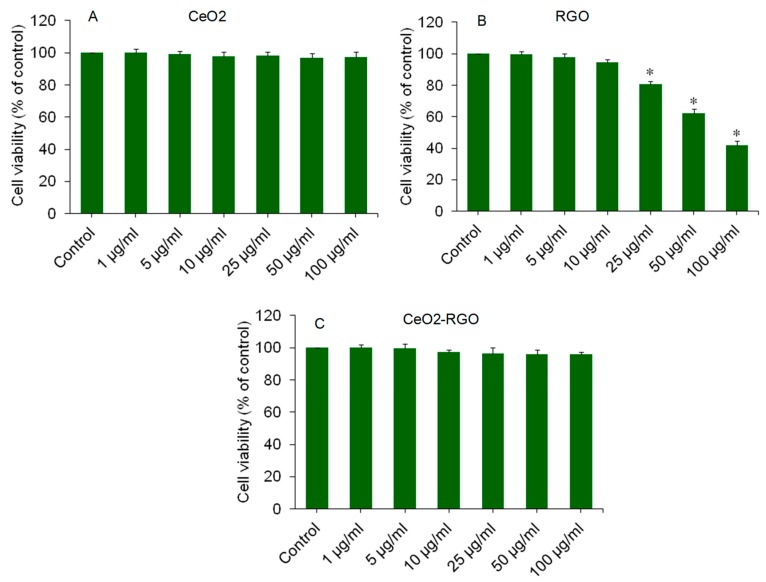
Cytotoxicity of CeO_2_ NPs, RGO, and CeO_2_-RGO nanocomposites in A549 cells. Cells were exposed for 24 h to various concentrations (1–100 µg/mL) of CeO_2_ NPs, RGO, and CeO_2_-RGO nanocomposites. (**A**) Cytotoxicity of A549 cells after CeO_2_ NP exposure. (**B**) Cytotoxicity of A549 cells after RGO exposure. (**C**) Cytotoxicity of A549 cells after CeO_2_-RGO nanocomposite exposure. Data are presented are mean ± SD of three independent experiments (*n* = 3). * Indicates a significant difference from the control group (*p* < 0.05). RGO: Reduced graphene oxide; CeO_2_: Cerium oxide; NPs: Nanoparticles.

**Figure 5 nanomaterials-09-01709-f005:**
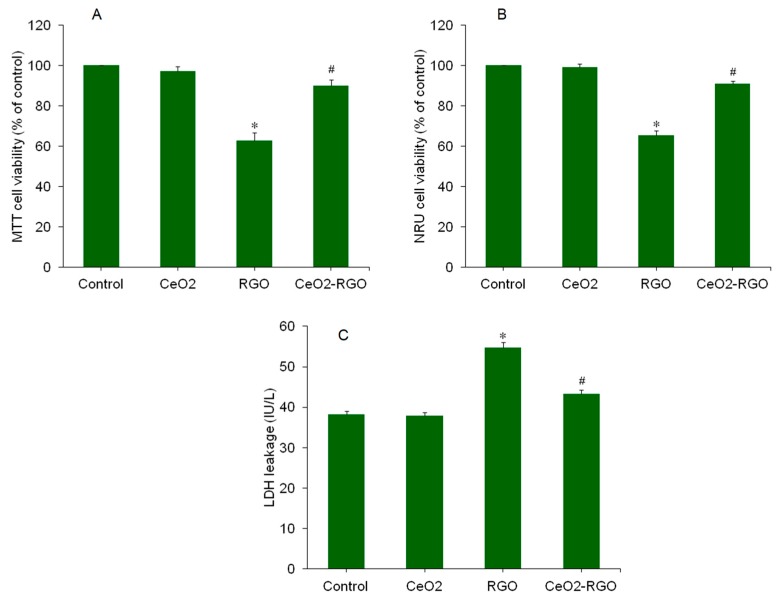
Cytotoxicity of A549 cells exposed for 24 h to CeO_2_ NPs (50 µg/mL), RGO (50 µg/mL), and a CeO_2_-RGO nanocomposite (50 µg/mL). (**A**) MTT assay. (**B**) NRU assay. (**C**) LDH leakage assay. Data are presented as mean ± SD of three independent experiments (*n* = 3). * Indicates a significant difference from control group (*p* < 0.05). # Indicates a significant difference from the RGO group (*p* < 0.05). RGO: Reduced graphene oxide; CeO_2_: Cerium oxide; NPs: Nanoparticles; NRU: Neutral red uptake; LDH: Lactate dehydrogenase.

**Figure 6 nanomaterials-09-01709-f006:**
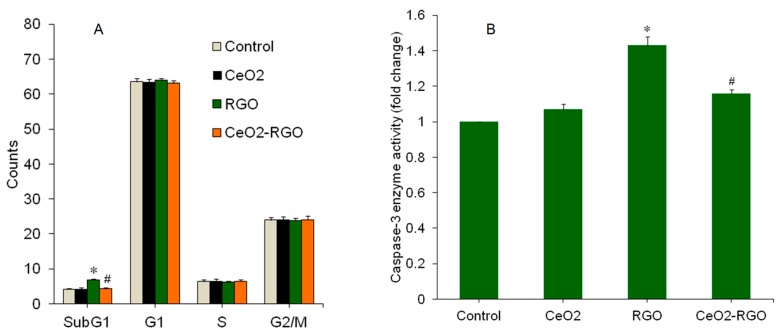
Analyses of cell cycle and caspase-3 enzyme activity of A549 cells exposed for 24 h to CeO_2_ NPs (50 µg/mL), RGO (50 µg/mL), and the CeO_2_-RGO nanocomposite (50 µg/mL). (**A**) Cell cycle analysis. (**B**) Caspase-3 enzyme activity. Data are presented as mean ± SD of three independent experiments (*n* = 3). * Indicates a significant difference from the control group (*p* < 0.05). # Indicates a significant difference from the RGO group (*p* < 0.05). RGO: Reduced graphene oxide; CeO_2_: Cerium oxide; NPs: Nanoparticles.

**Figure 7 nanomaterials-09-01709-f007:**
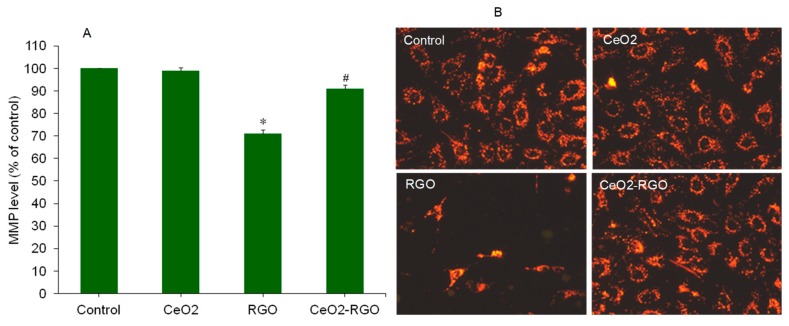
Mitochondrial membrane potential (MMP) level in A549 cells exposed for 24 h to the CeO_2_ NPs (50 µg/mL), RGO (50 µg/mL), and CeO_2_-RGO nanocomposite (50 µg/mL). (**A**) Quantitative data of MMP. (**B**) Fluorescent microscopy data of MMP. Quantitative data are presented as mean ± SD of three independent experiments (*n* = 3). * Indicates a significant difference from the control group (*p* < 0.05). # Indicates a significant difference from the RGO group (*p* < 0.05). RGO: Reduced graphene oxide; CeO_2_: Cerium oxide, NPs: Nanoparticles.

**Figure 8 nanomaterials-09-01709-f008:**
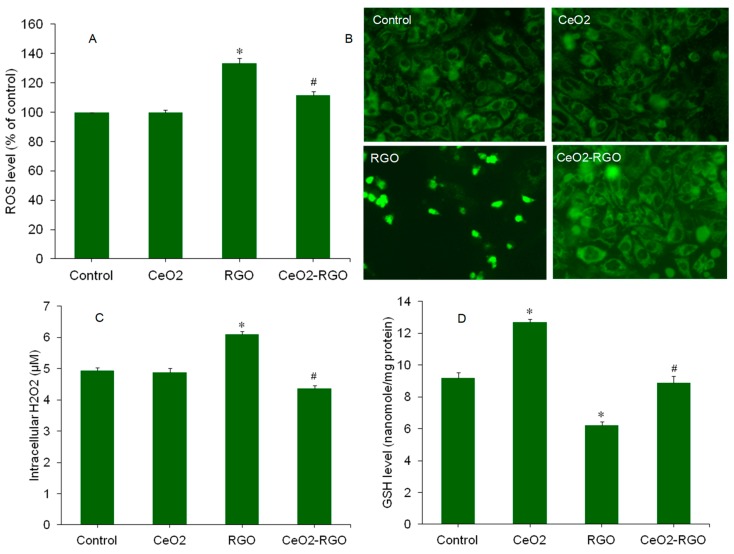
Oxidative stress response of A549 cells exposed for 24 to CeO_2_ NPs (50 µg/mL), RGO (50 µg/mL), and CeO_2_-RGO nanocomposite (50 µg/mL). (**A**) Quantitative data of the ROS level. (**B**) Fluorescent microscopy data of the ROS level. (**C**) H_2_O_2_ level. (**D**) GSH level. Quantitative data are presented as mean ± SD of three independent experiments (*n* = 3). * Indicates a significant difference from the control group (*p* < 0.05). # Indicates a significant difference from the RGO group (*p* < 0.05). RGO: Reduced graphene oxide; CeO_2_: Cerium oxide; NPs: Nanoparticles; ROS: Reactive oxygen species; H_2_O_2_: Hydrogen peroxide; GSH: Glutathione.

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
