# Peer review of "Evaluation of the Cytotoxicity and Oxidative Stress Response of CeO2-RGO Nanocomposites in Human Lung Epithelial A549 Cells"

_nanomaterials, 2019, doi:10.3390/nano9121709_

Round 1

Reviewer 1 Report

In the present work Ahamed et al. describe the toxicity of the graphene derivate Reduced Graphene Oxide (RGO) coupled or not to cerium oxide nanoparticles (CeO2-RGO), in the human lung adenocarcinoma cell line A549. They measure the cytotoxicity and the induction of apoptosis and oxidative stress by these materials (including CeO2 nanoparticles alone) 24h after exposure at a concentration of 50ug/ml.

Authors show that coupling CeO2 NPs to RGO reduces cytotoxicity and the induction of apoptosis and oxidative stress associated to RGO exposure alone. Authors propose that ‘CeO2 NPs converts cytotoxic RGO nanosheets into biocompatible CeO2-RGO graphene derivates’. The data are interesting and will help to design safe nano-materials for future applications in human health. Nanocomposites used in the present study are well-characterized and data are well presented. However, the are several issues to be addressed concerning the underlying mechanisms proposed according to the data and the methodology used.

Major points

Methodology.

1.Cell line. Authors select A549 cell line because they claim that is currently used for toxicology effects (refs 25,26). There are any other reasons to use a lung epithelial cell line? Is related to an eventual use of these nanocomposites during lung exposure? Concerning the use of  only one cell line along the study, as authors extensively described along the paper, toxicological data concerning nanomaterials could be controversial due to different models used to evaluate cytotoxicity. Cell lines could differ between each other and of course data obtained in cell lines could not represent more physiological models. Authors should perform some experiments in others cell lines and more important, representative experiments should be performed in relevant primary cells.

2.Time and doses of exposure. Why authors selected 24h? they are focusing in acute exposure? It is not justified. Concerning the doses, as authors show in Fig4 RGO toxicity is dose dependent, starting at 25ug/ml. Authors select 50ug/ml for exposure the cells to CeO2, RGO or CeO2-RGO, but 50ug/ml of CeO2-RGO is equivalent to 50ug/ml of RGO alone in terms of RGO content? Please clarify these points.

3.Mechanisms of action. Paper should intend to decipher potential mechanisms of action explaining CeO2-induced protection against RGO cytotoxicity. For instance, is coupling CeO2/RGO necessary? which is the effect of the addition of CeO2 NP and RGO at the same time (without coupling) in terms of RGO toxicity? In line with this, if cells are first preincubated with CeO2 NPs (favouring internalization?), are CeO2 NPs still protective against RGO toxicity?. Another important issue, is biodistribution of RGO the same that RGO coupled to CeO2 NP?, are they internalized in the same extend and are localized in the same cellular compartment? Protection induced by CeO2 is specific of CeO2 or other metallic NP (e.g. gold NPs) could also be efficient?

Minor points

- Lines 290-291.  Authors compare their data with those published by Qiang et al. (Pb2+ reduce cytotoxicity of GO through regulation of GO morphology). Have authors evidences about if this also happens in their model? Please clarify this comparison.

- Increases of ROS, intracellular H2O2 and caspase-3 activity are modest. Authors should include a positive control (a known inductor of ROS, H2O2 and apoptosis) in order to estimate the maximal threshold of the method used.

-Fig 8. Some panels are very poor quality (e.g. GO). In GO the probe goes to the nuclei? What does it mean? Please supply images of better quality and with nuclei stained.

-Please indicate if exposure to materials is performed in the presence of serum

Reviewer 2 Report

The manuscript entitled "Evaluation of cytotoxicity and oxidative stress response of CeO2-RGO nanocomposites in human lung epithelial A549 cells" describes the cytotoxicity of reduced graphene oxide compared to CeO2-RGO nanocomposites on A549 cells.

This paper lacks key in-depth discussions to attract readers' attention.

The potential therapeutic applications and scientific benefits of the results are not obvious and should be more emphasized.

The discrepancies between some published papers and the present results should be more analysed to write strong hypothesis and suggest leads to answers.

Please correct Dunnett (line 132).

Round 2

Reviewer 1 Report

No comments

Author Response

There is no comments from Reviewer during second revision.

Reviewer 2 Report

Despite author's statement related to the improvement of the discussion, it's hard to see any significant change in the discussion section that would help the reader to better understand the interest of this paper. The authors write "We have now discussed our results with emphasis of their application in biomedicine such as targeted drug delivery". Could the authors be more precised and specify the line number where the change was made? Again, the authors say they have discussed about "discrepancies between some earlier published work and their results in more detail". Could the authors specify where they added discussion elements?

The authors didn't precisely address any of my concerns.

Round 3

Reviewer 2 Report

English can be improved